# Building Green Smart City Capabilities in South Sumatra, Indonesia

**Hasmawaty [1],\*, Yulis Tyagita Utami [2] and Darius Antoni [3],\***

[1]   Faculty of Engineering and Computer Science, Baturaja University, Baturaja 32115, Indonesia
[2]   Communication and Informatics Office of South Sumatra Province, Palembang 30136, Indonesia; gitaummufatih12@gmail.com
[3]   Faculty of Engineering and Computer Science, Bina Darma University, Palembang 30452, Indonesia
\*   Correspondence: hasmawaty@binadarma.ac.id (H.); darius.antoni@binadarma.ac.id (D.A.)

**Abstract:** Information and communication technology is currently developing rapidly. It has created a great number of opportunities for governments worldwide to meet the demands of the public in providing services including environmentally friendly public services for the community and citizens. This study aims to present a case study on public services of South Sumatra government programs in order to develop a green smart city concept by combining smart city and green IT concepts which aim to align and incorporate green IT components, including pollution prevention, product stewardship, and clean technology into conceptual variants, essential elements, and strategic principles. A smart city is an innovative and modern concept utilising technology to facilitate and provide public information and to improve environmentally friendly public services through smart applications. South Sumatera is one of the provinces in Indonesia that has been implementing a smart city to provide environmentally friendly public services through the command centre. It is used to monitor an entire government agency's activities and communicates with the public. To fulfil the aims of the study, this study identifies the comprehensive environmentally friendly public services through Natural Resource-Based View Theory perspectives. In addition, this study deploys in-depth interviews with sources or informants as a form of data search and direct observation. The number of informants used as research respondents consisted of several elements of the government, including the executive, legislative, and implementing agencies. The study reveals that the South Sumatera Government has several environmentally friendly public services that could be deployed to prevent pollution and reduce the emission in their activities. Further, the government in South Sumatra has product and service stewardships and invests in clean technology to minimise the adverse impacts of their activities on the environment.

**Keywords:** smart city; Natural Resources-Based View Theory; South Sumatra; ecological; green IT

## 1. Introduction

In this era of industry 4.0 and globalisation, there is a rapid growth of interest in the public's, companies', and government agencies' use of Information Technology (IT) through information systems. This trend occurs because IT can process, store, and convert data into information needed by those parties [1]. In the governmental sphere, authorities must meet the demands of the public in providing services that are beneficial to the wider community in all regions; trusted and dependable; and accessible and interactive [2]. Information systems implementation is successful when users utilise the systems and such applications eventually support organisations in achieving their goals. For governments to implement information systems effectively and efficiently for public services, they must develop and build a management system such as a smart city strategy.

Indonesia Smart (Smart Nation) will only materialise when it has implemented the stages and development of a smart city at the city, district, and provincial levels. The

smart province itself is the concept of implementing governance at the provincial level based on digital/electronic where cities/regencies under the province have implemented smart cities and even smart villages in their public services. This shows that a smart province eventuates when all components in it, starting from the city/district levels to the village/complex areas, can collaborate. For this reason, proper communication planning and strategies are needed so that the implementation can perform optimally.

Responding to these challenges, the South Sumatra Provincial Government in November 2017 provided a legitimation in the form of Governor Regulation No. 47 of 2017 concerning the implementation of the South Sumatra Smart Province. This policy is an effort to utilise technology and IT that can support the Smart Indonesia strategy in the digital 4.0 era. The scope of this strategy includes planning, implementation, maintenance, and development, as well as monitoring of the South Sumatra Smart Province which harmonises and integrates the implementation of the South Sumatra Smart Province in each Regional Government Organisation (RGO) of the regency/city and provincial government with the central government information system. In terms of planning, as stipulated in the Governor Regulation No. 47 of 2017, this refers to the strategy of accelerating the use of information and technology through the implementation of the South Sumatra Smart Province service and the development of infrastructure and applications. The South Sumatra Provincial Government has appointed the South Sumatra Province Communication and Information Office as the service provider/project leader, and the manager of infrastructure and applications.

In developing a smart city, provincial, city, and district governments can implement distinct methods and approaches depending on their vision and mission. Preliminary research of Mora and Deakin [3] used information, communication, and technology (ICT) innovations for a sustainability approach to encourage environmentally friendly waste management and reduction with a mobile-based e-waste information system development strategy. In another study, Duan and Nasiri [4] employed a people, society, and technology approach. Evans and Karvonen [5] utilised a green information technology approach to developing an environmentally friendly-based smart city strategy. Xie and Tang [6] have successfully used the features of blockchain to improve smart city services and promote smart city development. The latter research also examined how blockchain technology is applied to smart cities in the world. This study will apply a green IT and environmentally friendly approach to developing a smart city in the province of South Sumatra.

Similar to many other developing countries, the government of Indonesia acknowledges the importance of green smart city in making public services more environmentally friendly to the public. Larasati and Handayaningsih [7] developed the green smart city concept based on the governance, branding, economy, living, society, and environment perspectives in Yogyakarta. Effendi and Syukri [8] developed the Nusantara sustainable smart city concept that consists of academia, business, community, government, and media. Tan and Taeihagh [9] reveal that the green smart city in developing countries including Indonesia can only be realised when concurrent socioeconomic, human, legal, and regulatory reforms are instituted. Sutriadi [10] states that a green smart city can be determined by community integrity, history and cultures, economic sectors, technology readiness and impacts, technical and political processes, and stakeholders. All these studies show that the development of a green smart city from different perspectives is needed in Indonesia. Despite the rapid development of the green smart city in Indonesia, the identification of IT capabilities in a smart city to provide environmentally sustainable government services still receives less attention. Therefore, there is a need to develop the green smart city based on IT capability to help the South Sumatra government to identify their IT resources that can be used to control and reduce emissions and wasted pollution during their activities.

Knowledge of these different methods and approaches is consequential when initiating implementation of the competence of green information technology (Green IT) to create environmentally friendly cities. Green IT in a smart city has the potential to make infrastructure services more efficient and reactive to user behaviour, minimise resource

consumption, improve environmental quality, and reduce $CO_2$ carbon emissions. It is this convergence of a 'Smart City' with urban sustainability that is the starting point for these issues.

Although the emergence of a green smart city approach puts digital innovation, a digital economy, and urban growth at the centre of efforts to create sustainable cities [11], how effective the use of IT capabilities has been for providing environmental sustainability services of government is unclear.

This study observes and identifies research gaps in the current smart city literature in Indonesia. Considering the lacunae, this study will develop a green smart city concept by combining the smart city and green IT concepts which aim to align and incorporate green IT components into conceptual variants, essential elements, and strategic principles. Therefore, this research ensures that social interests and environmental concerns have a special place in the smart city strategy. This green smart city initiative is used as a descriptive case study and chosen because of its ability to utilise ICT solutions to support the Indonesian government in fulfilling the transformative commitments that have recently been established in increasing public service innovations that have a characteristic concern for environmental damages.

## 2. Literature Review

### 2.1. Smart City Concept

Along with the development of information technology and the various types of services that governments can provide, the implementation of an e-government is often associated with the development of urban communities to create a technology-based city, known as the smart city concept [12]. In general, e-government is defined as the use of ICT by government agencies in conducting their duties to manage administration and provide public services [13]. Through ICT, the government can improve performance, bureaucratic efficiency, and quality of public services.

Regarding the smart city, it predominantly refers to city management using ICT to improve the economy, quality of life, and natural resource management through good and participatory governance [14]. The smart city concept is not only the ability to automate activities such as in public services or traffic systems, but also the ability to monitor, understand, analyse, and design a city management system to improve efficiency, social equality, and quality of people's lives in real-time [15]. To achieve this, smart cities apply and utilise ICT in various sectors in urban areas such as health services, energy, water, transportation, and waste management. According to Caragliu and Del Bo [16], there are several characteristics of a smart city that distinguish it from its conventional counterparts. They are development based on economic growth; utilisation of technology infrastructure and computer networks; increased role of the technology industry and creative industry in development; community participation in the implementation of development and public services; and sustainable natural resource and environmental management. In other words, smart city concepts focus on improving the quality of public services.

There are many studies that investigate the smart city and different perspectives to improve public service performance including technology, knowledge, and economic development. In terms of ICT infrastructure, many prior studies focus on the application of a wide range of electronic and digital technologies infrastructure to build smart cities. For example, Anttiroiko and Komninos [17] identify the use of ICT infrastructure to adjust how the citizen lives and works. Additionally, Ismagilova and Hughes [18] study the role of ICT infrastructure and citizens in order to create innovative and smart cities. Yaqoob and Hashem [19] investigate how the citizen could communicate and interact with the government through embedding information technology in smart cities. Similarly, Anttiroiko and Komninos [17] are concerned with developing ecosystem applications for public services as a strategic tool service of transformation of governments to meet the needs of citizens.

In terms of economic development, it has been associated with the existence of industries in the field of ICT or employing ICT in production processes. Moreover, the

economic approach could also attract creative citizens or people and encourage a city to be an economic hub through exploring the potential resources to improve the quality of living. For example, building high-quality and more efficient public transport that is used to connect labour with employment is considered an aspect for city growth. Thus, numerous cities in Southeast Asia attempted to develop and design a smart city project to promote economic growth through a smart city. In Singapore, this project is called The Intelligent Island, with information technology transforming work, life, and play. Other cities such as Taiwan are building e-Taoyuan and u-Taoyuan for improving quality of living and creating e-governance and ubiquitous possibilities.

Focusing on knowledge, a knowledge city widely adopts a smart city concept to nurture and empower citizens to deal with technologies. Winters [20] clarifies that a smart city is a centre of higher education, better-educated individuals, and skilled workforces. Smart cities act as magnets for creative people and workers, and this allows for the creation of a virtuous circle making them smarter and smarter. Consequently, a smart city has multiple opportunities to exploit its human potential and promote a creative life [21]. Glaeser and Berry [22] showed that the most rapid urban growth rates have been achieved in cities where a high share of the educated labour force is available. The buzz concept of being clever, smart, skillful, creative, networked, connected, and competitive becomes a key ingredient of knowledge-based urban development [3,4]. Possible confusion related to the technology perspective of a smart city comes from top-down and company-driven actions taken for creating a smart city. However, it also comes from the confusion with other similar terms, such as digital, intelligent, virtual, or ubiquitous city. These terms refer to more specific and less inclusive levels of a city, so that the concepts of smart cities often include them [16,23,24]. For example, a digital city refers to "a connected community that combines broadband communications infrastructure to meet the needs of governments, citizens, and businesses" [25]. The final goal of a digital city is to create an environment for information sharing, collaboration, interoperability, and seamless experiences anywhere in the city.

Indonesia Smart (Smart Nation) will only materialise when it has implemented the stages and development of a smart city at the city, district, and provincial levels. A smart province itself is the concept of implementing governance at the provincial level based on digital/electronic where cities/regencies under the province have implemented a smart city and even a smart village in its public services. This shows that a smart province eventuates when all components in it, starting from the city/district levels to the village/complex areas, can collaborate. For this reason, proper communication planning and strategy is needed so that the implementation can perform optimally. Prof. Suhono Harso Supangkat, in his paper titled "Smart Province" on 22 February 2018, described that the challenges faced in creating a smart province are: (a) equitable development; (b) human resource development to become smart people to play an active role in the provincial development process; (c) potential regional development; (d) the need for accurate and up-to-date information regarding regional conditions and development in all provinces (cities and districts); (e) participation of various stakeholders for the success of development; (f) the development of a digital government to assist in the process of data management and integration to support targeted policy-making and utilising ICT in potential development and governance in the province.

Although numerous cities are planned around the concept of sustainable economic development, knowledge and ICT Infrastructure, these cities promise to be strongly grounded in economic issues. Several people look at this as an example of a free-economic high-tech market in an area connecting developing and developed countries. However, most resources are consumed in cities worldwide, contributing to their economic importance, but also to their poor environmental performance. Cities consume between 60 percent and 80 percent of energy worldwide and are responsible for large shares of $CO_2$ emissions [26].

Thus, there is a need to develop a concept for an ecologically sustainable smart city as a new approach related to the role of information technology as a solution to reduce

the impact of smart city energy consumption. Table 1 shows the summary of the literature review on smart cities.

**Table 1.** Summary of literature on smart city.

| Authors | Pollution Prevention | Product Stewardship | Clean Technology | Economic Development | Infrastructure | Knowledge City |
|---|---|---|---|---|---|---|
| Mahizhnan [27] | | | | √ | √ | |
| Eger [28] | | | | √ | √ | √ |
| Giffinger and Gudrun [29] | | | | √ | √ | |
| Harrison, Eckman [30] | | | | | √ | √ |
| Thuzar [31] | | √ | | √ | | |
| Barrionuevo, Berrone [32] | | | √ | | | |
| Bakıcı, Almirall [33] | √ | | | | | |
| Popa, Carutasu [34] | | √ | √ | | | |
| Kumar and Dahiya [35] | | | | √ | | |
| Peng, Bohong [36] | √ | | | | √ | |
| Mwaniki, Kinyanjui [37] | | | | √ | | |
| Li [38] | | | | | √ | |
| Serrano [39] | | | | | √ | |
| Jnr, Majid [40] | | √ | | | | |
| Fromhold-Eisebith and Eisebith [41] | √ | | | √ | | |
| Kuecker and Hartley [42] | | | | | | √ |
| D'Aniello, Gaeta [43] | | | | | √ | √ |
| Yigitcanlar, Kankanamge [44] | | | | | √ | |

## 2.2. Green Smart City in Indonesia

Indonesia is a developing country which is attempting to implement a green smart city for efficiently and effectively adopting the latest technologies for improving the delivery of public services. This could be shown by several previous studies. For example, Mahesa and Yudoko [45] developed platform ecosystems for Indonesia smart cities that will be used to increase collaboration among stakeholders and develop new opportunities for the development of circular economy to resolve urbanization complexities. Larasati and Handayaningsih [7] identify dimensions of a smart city application to better understanding what constitutes a smart city and develop a concept to provide the general foundation for further smart city development. The dimensions of a smart city in this study are governance, branding, economy, living, society, and the environment. Achmad and Nugroho [46] develop a conceptual framework of the green smart city based on a synthesized and aggregated literature review. It shows that a green smart city framework is about integrating existing government services and resources (Table 2).

**Table 2.** Previous studies on green smart cities in Indonesia.

| No | Dimensions | References |
|---|---|---|
| 1. | Governance, Branding, Economy, Living, Society, and Environment | Larasati, Handayaningsih [7] |
| 2. | Tourism, Health, Safety and Security, Government, Energy, Environmental, Circular Economy, and Education | Mahesa, Yudoko [45], Kurniawan, Dwiyanto [47] |
| 3. | Services, Resources, Architecture, and Goals | Achmad, Nugroho [46] |
| 4. | Academic, Business, Government, Community Partners, and Media | Effendi, Syukri [8] |
| 5. | Environment Regulations, Availability of Green Spaces, Pollution, Investment, and Energy Efficiency | Afrianto and Tamnge [48] |
| 6. | Human Resources Capability | Rachmawati [49] |
| 7. | Social and Political Approach | Suartika and Cuthbert [50] |
| 8. | Environment, Social, Culture, and Economic | Hayati, Utami [51] |

The previous research findings on the implementation of a sustainable smart city focuses on governance, business, government, environment, energy, regulation, human resources, and economic, social, and political approaches. There is limited research on sustainable smart cities, which is related to identifying the use of information and communication technology in local governments to reduce its impact to the natural environment. This research reveals that ICT use enables the government to achieve their goals and maximize the performance of smart sustainable city services. A city needs to choose its role based on how it can optimally promote the development into a smart city while ensuring good strategic flexibility going forward. Therefore, there is a need for the Indonesian government, especially the South Sumatra government, to identify their ICT resources and capabilities in order to develop and provide sustainable government services to all citizens and the community.

### 2.3. Green Smart City Concept Development

To develop a green smart city concept, this study adopts the Natural Resources-Based View (Natural-RBV) theory. The theory aims to add the natural environment as a unique resource or ability to the RBV to develop a theory [52]. Hart argues that private or government organisations need to develop critical competencies in their engagement with the natural environment. Such competencies can contribute to sustainable competitive advantage. Therefore, Natural-RBV is developed through the relationship between environmental challenges and the resources of government and operationalised through environmental competence in the organisation.

This study adopts Natural-RBV theory to identify green IT capabilities that need to be studied and developed by the provincial government, especially communications and information departments in implementing green smart cities in South Sumatra. For example, when creating purchase, leasing, or outsourcing decisions, many local governments now consider companies that have a good environmental track record [53,54]. In addition, green IT functions are easily upgraded to meet business demands. It is also can educate employees and change their behaviour to reduce energy consumption. Therefore, green IT adoption is needed to help both private and government organisations build and improve their environmental competence.

Natural-RBV theory reveals that organisations can attain environmental competence through three factors [52,55]. The first factor is pollution prevention. This focuses on the control and reduction of emissions and waste pollution generated by the activities of government organisations. An organisation can enhance pollution prevention through improved management, material replacement, reuse, recycling, or process innovation [52,56]. Pollution prevention and waste management are accepted as part of the sustainability criteria of an organisation [57], and provide several advantages, especially for first-mover organisations (Nerht, 1996); they can also reduce cycle times by simplifying or eliminating unnecessary stages in an organisation's activities to increase productivity and efficiency (Hart 1995). Therefore, essentially, this factor is a strategic proposition (Elliot, 2007). Since the generation of pollution is considered a sign of inefficiency (Porter & Van der Linde, 1995), preventing pollution can enable organisations including governments to save control costs, input, and energy consumption, and reuse materials through recycling (Hart, 1997; Taylor, 1992). This can eventually increase the profitability of the organisation and its competitive position in the market (Hart 1995, 1997; Hart & Milstein, 2003; Molina-Azorin et al., 2009). Bakıcı and Almirall [33] reveal that electronic or digital services of a government can eliminate complicated procedures and reduce energy consumption through connecting people, information, and city elements using the new technologies.

The second factor is product stewardship. This refers to an organisation's ability to evaluate the environmental impact of its resources or infrastructure and services provided to partners or stakeholders. It requires environmental impacts to be considered throughout the entire lifecycle of the organisation, including the source of raw materials, product design, and the development process (Hart 1995, 1997). Product stewardship aims to

reduce the overall lifecycle environmental costs of a product by disciplining the design and development process to achieve system transformation from "cradle-to-grave" to "cradle-to-cradle" (Shrivastava et al. 1995). From a product design perspective, product stewardship can be considered a significant motivator for "green" application design that focuses on reducing energy over the full equipment cycle (Francis & Richardson, 2008). Product stewardship can also be enforced through laws and regulations. For example, a smart city policy leads to the sustainability of cities, particularly in the case of UK cities, to deal with their current and future development challenges and focus on city smartness and sustainability aspects [58].

The last one is clean technology. Such is an organisational strategy to invest in environmentally friendly technology to change organisational behaviour to be more sustainable (Hart 1995; Hart & Milstein 2003). It requires investment in future technology. Hart (1997) argues that organisations can reduce their unsustainable practices by planning, developing, and using clean technology. This is because many of the existing technology bases in many industries are non-environmentally friendly. At this stage, organisations need to use their sustainability vision to plan for new products and services they should develop or purchase, and the capabilities and competencies that will be required to use them for more sustainable options. Cleaner production used to achieve eco-sustainability especially in the production process can allow industrial production to enter into the environmental sustainability vision of Hart and Milstein (2003), who highlighted the potential use of technology that saves materials, is energy-efficient, non-polluting, and low waste (Hart, 1997; Geiser, 2001).

In the case of a green smart city, clean technology can be used for tackling pollution, managing water efficiently, and supporting green buildings and alternative energy; as a result, cities can become cleaner, more pleasant places to live, while at the same time drastically reducing their energy bills [32]. For example, the government of Moncton, Canada, has shown a broad green city project, which has the aim of promoting greater use of public transport and bicycles, and more recycling. In addition, with ICT, the government has imposed tighter controls on irrigation water and launched tree-planting campaigns [32]. In the same way, Popa and Carutasu [34] state that by using ICT infrastructure, including the Internet of Things (IoT), governments can optimise the use of smart waste systems to monitor and track waste collection in order to improve its productivity and collected waste storage capacity. Moreover, cloud-based technology for data-driven water demand management utilises data analytic methodology to optimise water-use efficiency and improve financial forecasting accuracy by engaging citizens [59]. This technology can be used as a software-as-a-service application that allows cities access to both real-time and historical parking data and aims to make optimal and efficient use of parking resources.

## 3. Research Methodology

This study employed a qualitative research and focus group method based on the philosophy of post-positivism, which was used to examine the natural conditions of the object, and the researchers were the principal instrument. The sampling of data sources was conducted purposively, the collection technique was managed by triangulation (combined), the data analysis was inductive, and the results emphasised meaning rather than generalisation.

The object of this research was the Communication and Informatics Office of South Sumatra Province, Related RGO, District/City Information and Communication Services, providers, and universities interested in efforts to create a green smart province. This study deployed in-depth interviews with sources or informants as a form of data search and direct observation. The number of informants used as research respondents consisted of several elements of the government, including the executive, legislative, and implementing agencies as shown in Table 3. They were selected for interviews based on the role in their respective organisations and who is involved in the strategic planning process during building and implementing the smart city. The questions the researchers asked the informants

varied according to the capacities of the informants. The interviews were exploratory in nature, consisting of open-ended questions that focused on participant perceptions of the processes and influential factors of green IT in the South Sumatra government with respect to the three dimensions of natural RBV Theory. In Indonesian Ministry of Communication and Informatics cases, the interviews focused on their perceptions of these issues in IT projects implemented to reduce energy consumption in operational activities and clearly defined the roles, responsibilities, accountability, and control for a green smart city. An informant from the department of communications and informatics of South Sumatra was questioned regarding the development of green IT standards across government agencies and offices and the extent to which the South Sumatra government has a green business infrastructure (such as green rated buildings) and green power sources. Furthermore, the Regional House of People's Representatives of South Sumatra emphasized allocation of budgetary and other resources for Green IT and the extent of policies in the house dedicated to e-government including green smart cities.

**Table 3.** Research informants.

| Informant | Total | Informant Status |
|-----------|-------|------------------|
| Indonesian Ministry of Communication and Information | 1 | Researcher at the Indonesian Ministry of Communication and Information Research and Development Centre |
| Regional Secretariat of South Sumatra | 2 | Regional Secretary and Assistant III for Administration and General Affairs |
| Regional House of People's Representatives of South Sumatra | 1 | Chairman of Commission I |
| Department of Communication and Informatics of South Sumatra | 7 | Head of Service, Secretary, Head of PIP, Head of E-Gov, Head of ICT and Encoding, Head of Statistics and Head of Planning |
| Total | 11 | |

In order for this research to be more objective and accurate, researchers also sought additional information by making field observations. Table 3 shows the details of the informants in this study. Analysis of the data in this study was communication planning model based on Public Relations by Cultic and Centre to identify and analyse the communication strategies managed by the South Sumatra Province Communication and Information Office in realising South Sumatra as a smart province in implementing the Sumsel Command.

The second phase is focus group methodology. This method is well-documented as a reliable and cost-effective method for qualitative data gathering in both public and private organisations. In total, 15 participants in a focus group are involved in the process due to a shared government circumstance or condition. This method also has the strength of a focus group in providing insights into a specific issue from a group of selected participants. As the focus group technique relies on effective interaction between several participants, well-designed focus groups provide the researcher with the ability to observe how theories emerge with regard to the viewpoint of the participants. In this respect, it is noted that the opportunity must be offered to all participants to express their thoughts. This study provides a deeper understanding on the three key dimensions and expected data outcomes from a focus group: (1) articulated data, where participants express thoughts from a direct question; (2) attributional data, where the moderator discreetly provokes discussion; and (3) emergent data, which refers to normative understandings.

## 4. Research Findings and Discussion

Based on the results of interview and previous studies, this study proposes a conceptual model consisting of three components corresponding to the three factors of IT use in

smart cities as follows: pollution prevention, product stewardship, and clean technology. This study discussed each component in more detail.

The research results for the development of a smart green city are highly needed and is one of the considerations in planning for the province of South Sumatra. Pre-development research is needed to determine the feasibility of the programme, whether it is effective, efficient, and trackable, whereas post-development research functions to determine whether the programme is acceptable. In the development and construction of a smart city in South Sumatra, the implementation of the Sumsel Command Centre was conducted by the South Sumatra Province Communication and Information Agency on 7 September 2018 as a strategy to improve cost efficiency and operational time for the South Sumatra Provincial government. To develop and build a strategy from implementing the South Sumatra Command Centre, this study employed three eco-sustainability strategies as follows: pollution prevention, product stewardship, and clean technology (Ijab, 2010).

### 4.1. Pollution Prevention

In this strategy, several of the data and information the researcher requested of the informants included: the number of users of government public service applications, pollution and waste management, procedures for reuse, recycling, or the innovative process of using IT implemented by a smart province. According to the researcher, data is needed to take the appropriate steps in determining the strategy to be implemented by governmental agencies. Moreover, this programme will have a significant impact on making South Sumatra a smart and environmentally friendly province. This smart solution is obtained from comprehensive data and information through the use of ICT. This will help to develop the efficiency of government services to the community appropriately and quickly.

At the beginning of 2019, the research process began at the Human Resources Development and Research Centre (BPSDMP) in Jakarta by the HR Research and Development Agency of the Ministry of Communication and Informatics with the title: "Smart Province Readiness Study in South Sumatra Province". This research will continue this year by conducting a focus group discussion (FGD) on the concept of the ideal smart province and the readiness of the smart province. According to the head of the Human Resources Research and Development Agency of the Ministry of Communication and Information Technology of the Republic of Indonesia, when contacted by telephone (10/2), the results of the FGD conducted and attended by all discussion participants (all representatives of districts/cities in South Sumatra) suggested that the government must implement a strategy that takes into account the main potential of the region, as well as procedures for preparing technology (infrastructure, applications) to create a smart area, and prepare governance (legal basis, policies, SOP), sources of human resources, and budgets that will support South Sumatra as a smart province. Apart from that, the informant also mentioned that there is a need to build a relationship between the South Sumatra government and the Regional House of People's Representatives of South Sumatra (DPRD), which has substantial political influence in implementing environmental-based smart provinces. One example is the implementation of this command centre to support the 2018 Asian Games, an international event held in Palembang, South Sumatra.

The Chairman of Commission I in DPRD confirms that "as we know, the DPRD has several functions and tasks, one of them is the budget work, where the DPRD functions to discuss the draft regional regulation on the provincial APBD (regional budget). The DPRD also has a supervisory function, in this case the supervision of the regional government and also the supervision of the provincial APBD funds. It means if South Sumatra government implement something; the provincial government must have a good communication and relationship with the DPRD to implement green smart city".

The Department of Communication and Informatics of South Sumatra also states that "many IT human resources would disregard organisational objectives for eco-sustainability if they were not measured on this performance. Both the provincial and local govern-

ment indicated that they had clearly defined sustainability vision, associated policy and governance mechanisms that facilitate the development of Green IT policies".

The results from this study are consistent with several prior studies. Tolbert and Mossberger [60] state that the legislature is an important factor in determining whether provinces will innovate in digital government. This is because that presence of legislative committees in the House dedicated to e-government is critical in explaining the extent of policy innovation of digital government. Many countries with advanced IT infrastructure and legislative policy making capacity have more extensive implementation of e-government over time. Pang [61], in his study titled "The moderating effect of IT governance on the relationship between IT investments and government performance" shows the local government requires IT budget approval from the legislature as part of IT governance in public sector organisations.

In addition, the South Sumatra Government has to redesign their IT infrastructure ecosystem to reduce energy consumption. This finding is supported by [62,63] studies, which reveal that redesign IT infrastructure is required to meet environmental change through innovating business processes of e-governments. For example, the implementation of e-mail, digital signatures, and virtual technologies can help institutions to promote their green government initiatives. Similarly, Antoni and Jie [64] discovered that redesigning the IT infrastructure in business processes can be emphasized by reducing operational costs in an organisation. For example, the use of video conferences in command centres such as Zoom, Google meet, and other applications can improve efficiency of transportation cost, and might drive the government to become an environmentally friendly office. In addition, the IT infrastructure ecosystems have to accommodate stakeholders of government demands to perform environmentally friendly activities. It might be used as a strategy of the South Sumatra government to encourage internal and external stakeholders to adopt green behaviour in their activities, such as replacing the paper-based documents with digital documents. Furthermore, the IT infrastructure might reduce the digital divide and enhance citizen participation in e-government implementation [65]. It means that the people with low-income households, people living in informal settlements, and people with lower levels of electronic literacy are able to access the smart city apps and also communicate with local government through the provided IT infrastructure channels, including websites, mobile apps, and kiosks [66]. Accordingly, it can be concluded that South Sumatra Province has had several strategies to support the implementation of an environmentally friendly smart city, strategies that have either been realised or are useful for the future.

### 4.2. Product Stewardship

In terms of product stewardship, the Head of the Research and Development Agency for Human Resources for the Ministry of Communication and Information Technology of the Republic of Indonesia emphasised equitable development strategies, including developing human resources whose ability to use environmentally friendly technology, such as teleconferencing or video conference technology found in the command centre, is used as a means of interaction and virtual meetings with regional heads of all regencies and cities in South Sumatra and can also monitor all activities of all regional organisations. He also added that the ability to use technology is insufficient. It must be accompanied by the ability to manage infrastructure and redesign service procedures for the community to minimise natural damage caused by service activities. Based on the Strategic Plan of the South Sumatra Province Communication and Informatics Office for 2018–2023, it is stated that the authority of the South Sumatra Province Communication and Information Office is in the communication and informatics sector; as well, the encoding and statistics in processing its authority emphasises the dissemination of information throughout South Sumatra through the use of existing ICT infrastructures. This is in line with the implementation of a command centre which aims to create an environmentally friendly South Sumatra Smart Province through the use of ICT infrastructure, data management, and

applications that make it more straightforward for users and the public. South Sumatra Province Communication and Information Office makes this point clear:

> "Substituting travel and physical meetings with building command centre including videoconferencing and collaboration tools, eliminating paper-based workflows and reporting, and conducting government services through integrated electronic-based government systems including applications. In addition to these applications, other studies describe the indirect positive impact of using IS for reporting and measurement of environmental government services or collaborating on environmental initiatives."

Based on the results of interviews with the informants, this study found at least six green products or services from the South Sumatra Communication and Information Office. First, the implementation of the command centre as a central facility for controlling and monitoring the components of the South Sumatra Smart Province in the form of applications, data, and information owned by RGOs throughout South Sumatra. Currently, this space is connected to various web-based applications such as the official portal of South Sumatra go.id, Sumsel smart digitalisation, Indonesian Disaster Rapid Assessment, and Integrated Media Management.

This finding is consistent with the Huang [67] study that found that the role of the command centre is to be a nerve centre for efficiently and effectively mobilizing resources, coordinating human resources, and providing advice guidelines during the COVID-19 period. This current finding also agrees with the Lacity and Willcocks [68] study findings that a centralized command centre establishes standards and best practices and tracks the business performance of service automation which can reduce human intervention to a minimum and lead to enhanced information efficiency.

Second, the South Sumatra Communication and Informatics office is building data integration and applications from either RGO or cities and regencies in South Sumatra. This will transform the command centre into a data and information centre for RGO, cities, and regencies in South Sumatra. This finding is consistent with the Molla [69] study finding in that an advanced data centre provides services on how to position IT as an enabler of green initiatives. In addition, the finding of this study is also similar to the Hashem and Chang [70] study finding that integrated data promises flexibility and low costs to reduce the technical barriers of addressing the data. For example, integrated data collected from multiple sources, such as citizen and government organisations, are stored in a database. The data are able to be utilized by the business intelligence and data analytics model to predict future behaviour with increasing precision, decision automation, data driven business, and performance management to establish government administration simply. Thus, the green smart city infrastructures could be designed as platforms that are suitable for government and citizens to improve capacity and engage public participation [65,71].

Third, the implementation of closed-circuit television (CCTV) used as a support public service. The command centre operator ensures that the CCTV at a certain point is on and monitors the state of the area. The results subsequently will be analysed in more detail so that notifications appear as needed. This finding is consistent with the Chung [72] research finding that CCTV is combined with intelligent technology and the internet, which automatically detects and identifies specific objects such as people and objects. The implementation of the intelligent CCTV in the government office will be made mutually integrated with the employee access system. The data obtained by the data centre will be more detailed in monitoring employees while carrying out their work activities. Thus, by using intelligent CCTV, employee performance can be efficiently and effectively measured. As well as in this system, there is also information that can be used as an indicator of performance measurement when making a decision immediately.

### 4.3. Clean Technology

The last strategy is to create environmentally friendly products or services by using clean technology. The Communication and Informatics Office of the Province of South

Sumatra must continue to research to support the ideal smart province strategy for the success of the environmental-based smart province. This can be achieved through the online media approach that people need for communication (e.g., what information is needed, whether entertainment, opinions, or news). Further, existing data can be used to analyse communication components, starting from sources, messages, channels or media, recipients, and feedback from the public. In addition to conducting research, the South Sumatra Province Communication and Informatics Office can optimise the role of the Information and Documentation Management Officer (IDMO) Assistant of the South Sumatra Province Information and Information Technology Office in collecting public information related to the field of information and communication technology. In the future, concerning strategies to implement clean technology, governments in South Sumatra, especially the Communication and Information Technology Office, will invest in building services and products that are environmentally friendly, including building a data centre that provides data and information of policies for public services, and provision for information on local government administration. The Department of Communication and Informatics of South Sumatra make the point clearly:

> " . . . the data centre is a collection of data and information obtained through entire government services in the South Sumatra government, which will be used to improve the quality of existing services and be used to predict future sustainable services of government".

> " . . . we might employ our official website as a sustainability strategy to place all our Green smart city policies including general sustainability policy and the environmental management system is all seen in there too . . . "

This environmentally friendly strategy shows the government efforts to invest in data centres to meet stakeholder requirements to provide green products and services and build a green brand image that influences the mindset of stakeholders [55].

Secondly, the South Sumatra Government has a green initiative to develop and upgrade infrastructure to increase the accessibility of information and ICT resources for the public. This government's initiative can empower the stakeholders to access whole e-government channels from any location. Therefore, the stakeholder does not need to go to a government office to obtain a public service, which reduces the transportation cost. Thirdly, the Province of South Sumatra Government optimises the implementation of the electronic-based Government Administration System (EGAS) to provide environmentally friendly services. This program encourages the government to improve operational efficiency by implementing paperless offices [73]. This also complies with the requirements and targets of paperless initiatives in Indonesian public sector services.

Implementation of a public information disclosure of EGAS is in accordance with the mandate of Law No. 14 of 2008 and its implementing regulations. This finding is supported by the Dawes and Vidiasova [74] research finding that through implementation of regulations, the government has the guidance to keep on track to develop IT infrastructure, and the government itself receives benefits in terms of progress towards political and strategic goals for transparency, public service, and good management along with improvements in stewardship and agency mission accomplishment. Furthermore, the management systems standards and standardisation of IT equipment can assist the South Sumatra government as guidelines in minimising the environmental impacts of their public services. These also help the local government to select suppliers, raw materials, and products that can be used for green smart city infrastructure. Despite monitoring the results and undertaking an evaluation and inventory of the ministries about the implementation of government affairs in the field of ICT of South Sumatra for a period of two years, the existence of the main tasks and functions of ICT affairs are still scattered in various regional apparatuses (agency authority has not been optimally implemented). Table 4 shows there are the strategies for the South Sumatra Provincial Government in realising an environmentally friendly smart city.

**Table 4.** Ecological smart city.

| No | Ecological Sustainability | Public Services of Smart City |
|---|---|---|
| 1. | Pollution Prevention | SOP for infrastructure and applications<br>ICT ecosystems<br>Local regulation<br>Human Resources capabilities<br>IT Infrastructure Budget |
| 2. | Product Stewardship | Sumsel Command Centre<br>Virtual meetings<br>Monitoring RGO activities<br>Disseminating information through social media and websites<br>E-sumsel (integrated budget Application)<br>Data and application integration<br>CCTV implementation |
| 3. | Clean Technology | Data centre for public services<br>Accessibility of information for each city district<br>Optimisation of Electronic-Based Government Administration Systems<br>Local government electronic data and information management services<br>Increasing literacy in the use of digital start-ups by businesses/MSMEs in South Sumatra |

## 5. Conclusions and Limitations

This paper attempts to identify the concept of an ecological smart city, which is becoming increasingly popular in Indonesia. An in-depth analysis of interviews and focus groups reveals that the ecological smart city can be investigated from several perspectives, including pollution prevention, product stewardship, and clean technology. These three dimensions should be the guidance of the South Sumatra Government to identify the green smart city implementation. Based on the results of the analyses, pollution prevention is determined by the public services of a smart city, regional resource potential, technology preparation procedures for infrastructure and applications, ICT ecosystems, local regulations, improvement of human resource capabilities, coordination (both vertically and horizontally in data/information management activities), and IT infrastructure budgets. Moreover, this research also shows that product stewardship consists of Sumsel Command Centre, virtual meeting infrastructure, monitoring RGo activities, SOP of green design, disseminating information through social media and websites, E-sumsel as a budget integrated system, one data and application integration, and CCTV implementation. In the clean technology approach, the study reveals South Sumatra Province has several IT strategies and uses environmentally friendly IT resources in order to build images of green smart cities in Indonesia. These technologies include data centres, accessibility of information for each city district, optimisation of electronic-based government administration systems, local government electronic data and information management services, increasing literacy in the use of digital start-ups by businesses/MSMEs in South Sumatra, application of encryption for information security, implementation of public information disclosures, and results of monitoring and evaluation of environmentally friendly IT infrastructure policies.

In closing, it is important to acknowledge several limitations in this study and suggest possible paths for further research. First, the ability to generalize from the results of this study is limited by the small number of cases and interviews conducted. Secondly, this research only focuses on the development of green smart cities within the South Sumatra province. Thirdly, the current research has provided both ecological and smart city viewpoints. Fourth, arguably, research has identified three dimensions and 23 public services of green smart cities based on the ecological model developed by Hart [52]. While acknowledging this limitation, more work will be required to be carried out to develop a green smart city for other cities in Indonesia or other developing countries from different

perspectives and approaches. For example, the COVID-19 pandemic, which has driven more government public services move from offline to online, has provided new opportunities for implementing green smart city concepts in other provinces of Indonesia and other developing countries. Future research should also investigate a broader range of issues and approaches and ensure more provinces become involved to obtain more understanding of how a green smart city could be implemented, and might investigate the extent to which a green smart city can be implemented in South Sumatra Province or other areas to provide excellent services to citizens.

**Author Contributions:** The main research activities of the authors can be described as follows: Conceptualization, H., Y.T.U. and D.A.; methodology, Y.T.U. and H.; formal analysis, H. and D.A.; investigation, Y.T.U. and D.A.; writing—original draft preparation, Y.T.U. and D.A.; writing—review and editing, H. and D.A. All authors have read and agreed to the published version of the manuscript.

**Funding:** This research received no external funding.

**Institutional Review Board Statement:** Not applicable.

**Informed Consent Statement:** Informed consent was obtained from all subjects involved in the study.

**Data Availability Statement:** Not applicable.

**Conflicts of Interest:** The authors declare no conflict of interest.

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
