# Peer review of "Building Green Smart City Capabilities in South Sumatra, Indonesia"

_sustainability, doi:10.3390/su14137695_

Round 1
Reviewer 1 Report
Dear authors
The work is well-built and interesting. However, some areas need improvement.
- The concept of a green smart city is unclearly described, it needs more attention.
- Please check the introduction, it should be improved by considering the general idea to a specific case of an Indonesian city.
- Literature is succinct.
- Please consider these studies to improve the literature. a. http://dx.doi.org/10.1002/sd.2169 b. https://iiteda.org/wp-content/uploads/2022/01/Book-of-Abstracts-26-January-2021.pdf#page=13 3. https://www.tandfonline.com/doi/full/10.1080/10447318.2021.2012385
- Arguments are good but can not be generalized.
- Overall, the paper can be improved to build a logical story.
Author Response
Dear Reviewer,
Please find the revised paper attached.
Thank you

Reviewer 2 Report
Dear Colleagues,
Thank you for the opportunity to get acquainted with your research and its findings. Undoubtedly, it is of deep concern and relevance for the public, authorities and researchers in terms of territorial development.
However, the study entails in-depth systematization and validity of conclusions and proposals. In particular, an arrangement of interview questions for officials, being chosen as target respondents, is ambiguous and debatable. In our opinion, a survey of potential consumers of smart territory services represented by the population of different groups would be more informative.
The authors have proposed quite a lot of evident and currently known smart territory characteristics accustomed in other countries.
To systematize the proposals, the authors should have grounded on the current strategies for the development of the target area. Given the structure of the paper, it is advisable to make suggestions within each section of the strategy, thereby providing more systemic recommendations.
It should be noted that the authors have made an interesting attempt to categorize the smart factors of the ecological module, having paid special attention thereto. However, the boundaries between the three groups are blurred, and the fragmented proposals should be systematized.
Author Response
Dear Reviewer,
Please find revised paper attached
thank you
Darius Antoni

Reviewer 3 Report
This paper explores the concept of smart cities in the context of South Sumatera. The topic is interesting. Here are my comments to improve the paper:
1- the title of the paper is too generic. Try to come up with a title that is more informative about what your paper is about.
2- in the abstract, you need to include the major aspects of the entire paper in a prescribed sequence that includes: 1) the overall purpose of the study and the research problem(s) you investigated; 2) the basic design of the study; 3) major findings or trends found as a result of your analysis; and, 4) a brief summary of your interpretations and conclusions. At the moment, some of these aspects are missing.
3- In the introduction, make clearer what knowledge gaps you identified and how your research addresses them. Also, make the research objectives/questions clearer. Answer the “so what?” question. Why investigating such matter is important? End the introduction with an outline of the paper; what comes next?
4- The novelty/originality should be clearly justified that the manuscript contains sufficient contributions to the new body of knowledge from the international perspective. What new things (new theories, new methods, or new policies) can the paper contribute to the existing international literature? This point must be reasonably justified by a Literature Review, clearly introduced in Introduction Section, and completely discussed in Discussion Section.
5- It would be helpful to include some discussions of e-participation in your paper which concerns smart cities. In this discussion you can include both opportunities and challenges of employing e-participation. Here are some recent references:
https://doi.org/10.1016/j.cities.2021.103281
https://doi.org/10.4018/978-1-6684-3706-3.ch044
https://doi.org/10.1177/2399808317712515
6- in the methodology, include type of questions you asked the interviewees. Also, include some of the questions to give the reader a sense of your questionnaire.
7- What are the limitations of your study?
8- you need to include more direct quotations from the interviews in the result section to support your arguments.
9- you need to refer back to the literature and previous studies in your result, discussion and conclusion sections.
10- how generalisable your findings are other places? Provide some discussions around the generalisability of your findings in the discussion section.
Author Response
Dear Reviewer,
Please find revised paper attached
Thank you

Round 2
Reviewer 1 Report
Impressive work. It can be accepted for publication.
Author Response
Dear Editor,
Thank you for your help and make our paper better
Regard,
Darius Antoni
Reviewer 2 Report
Dear colleagues, thank you for your work. I think that you version 2 of your manuscript can be publish. Some remarks can be improve in the next researches.
Author Response

(The authors gave the same response as above.)

Reviewer 3 Report
Thank you for addressing my comments. The paper has been considerably improved. However, a few of the comments have not been fully addressed and I would like to give another opportunity to the authors to address the comments:
Comment 1: I still think the title is too generic. Perhaps replace 'developing country' with South Sumatera or Indonesia.
Comment 5: make sure you have included the recommended references to strengthen your arguments.
Comment 7: you need to further think about the limitations of your study.
Comment 10: similarly, you need to further think about the generalisability of your findings in the discussion section.
Author Response
Dear editors,
we do our best to revise our paper based on your comments.
we have replaced "developing country" with South Sumatra, Indonesia and we also use some recommended references and put them in our paper.
we also revise our paper based on Comments 7 and 10.
Thank You,
Darius Antoni
